# Evaluating a Universal Dependencies Conversion Pipeline for Icelandic

**Þórunn Arnardóttir[1], Hinrik Hafsteinsson[1], Atli Jasonarson[2],**
**Anton Karl Ingason[1], Steinþór Steingrímsson[2]**
[1]University of Iceland, [2]The Árni Magnússon Institute for Icelandic Studies
`{thar, hinhaf, antoni}@hi.is,`
`{atli.jasonarson, steinthor.steingrimsson}@arnastofnun.is`

## Abstract

We describe the evaluation and development of a rule-based treebank conversion tool, UDConverter, which converts treebanks from the constituency-based PPCHE annotation scheme to the dependency-based Universal Dependencies (UD) scheme. The tool has already been used in the production of three UD treebanks, although no formal evaluation of the tool has been carried out as of yet. By manually correcting new output files from the converter and comparing them to the raw output, we measured the labeled attachment score (LAS) and unlabeled attachment score (UAS) of the converted texts. We obtain an LAS of 82.87 and a UAS of 87.91. In comparison to other tools, UDConverter currently provides the best results in automatic UD treebank creation for Icelandic.

## 1 Introduction

The Universal Dependencies (UD) project is a multilingual project, consisting of dependency treebanks in 138 languages (Zeman et al., 2022; Nivre et al., 2020). UDConverter is a tool which converts a phrase structure treebank to a UD treebank (Arnardóttir et al., 2020), and has been used for creating three UD corpora. Originally configured for Icelandic, the converter can be extended to convert treebanks in languages other than Icelandic, as has been done for a Faroese treebank (Arnardóttir et al., 2020), but it has not been thoroughly evaluated until now. Without such evaluation, the benefit of using the converter is uncertain. Therefore, we manually corrected a portion of a treebank created with the converted UD treebank and evaluate the conversion by comparing the converted sentences' output to the manually corrected ones. The evaluation is used to guide further development of UDConverter, resulting in an improved conversion pipeline.

The paper is structured as follows. Section 2 discusses relevant resources, including UD corpora and methods of creating them. Section 3 describes the evaluation setup used while Section 4 discusses the results, including initial results before the converter was improved. We compare the converter's accuracy scores to the accuracy of three UD parsers in Section 5 and finally, we conclude in Section 6.

## 2 Background

UDConverter is a Python module for converting bracket-parsed treebanks in the format of the Penn Parsed Corpora of Historical English (PPCHE) to the Universal Dependencies framework (Arnardóttir et al., 2020). It was created in order to convert the Icelandic Parsed Historical Corpus (IcePaHC) (Rögnvaldsson et al., 2012) to the UD CoNLL-U format and has been used for creating three UD corpora, UD_Icelandic-IcePaHC, UD_Icelandic-Modern and UD_Faroese-FarPaHC, all included in version 2.11 of Universal Dependencies (Zeman et al., 2022). The converter takes an original IcePaHC-format tree and converts it to a UD tree, displayed in the CoNLL-U format. As discussed in Arnardóttir et al. (2020), the converter can be extended to convert treebanks in other languages than Icelandic, as long as the input treebanks are in a format similar to the IcePaHC one. The converter's output generally adheres to UD annotation guidelines but no formal evaluation of the converter has been carried out until now.

The UD corpora which were created by using UDConverter were all converted from pre-existing constituency treebanks. These treebanks were manually annotated according to the PPCHE annotation scheme (Kroch and Taylor, 2000; Kroch et al., 2004), which uses labeled bracketing in the same way as the Penn Treebank (Taylor et al., 2003).

This IcePaHC annotation scheme was used as a basis for the rule sets of UDConverter.

UD_Icelandic-Modern was converted from 21st-century additions to IcePaHC, consisting of modern Icelandic texts (Rúnarsson and Sigurðsson, 2020). It contains genres not previously found in the original IcePaHC (Wallenberg et al., 2011), extracted from the Icelandic Gigaword Corpus (Steingrímsson et al., 2018).

Two UD parsers have recently been released for Icelandic (Jasonarson et al., 2022a,b), both of which utilize information from a pre-trained BERT-like model, in this case an ELECTRA model that was pre-trained on Icelandic texts (Daðason and Loftsson, 2022). One of the models was trained with DiaParser (Attardi et al., 2021), an extended version of the Biaffine parser (Dozat and Manning, 2017), which uses contextualized embeddings, as well as attentions, from a transformer model as its input features. The other one was trained with COMBO (Klimaszewski and Wróblewska, 2021), which accepts pre-trained embeddings from a transformer, as well as character and lemma embeddings, in addition to part-of-speech tags, as its input features. Both parsers were trained on two Icelandic UD corpora, UD_Icelandic-IcePaHC and UD_Icelandic-Modern.

## 3 Evaluation

In order to evaluate UDConverter, we set up a testing experiment where output CoNLL-U files from the converter were manually evaluated and corrected per strict annotation guidelines. These were then compared to the original raw output files. As these files contain identical texts, this enabled a one-to-one comparison, with the manually corrected files serving as a gold standard.

In our evaluation, we focused on measuring the accuracy of the conversion when it comes to heads and dependency relations. For this project, we chose to source sentences for manual correction from the UD_Icelandic-Modern corpus, discussed in Section 2, which then became the test set. In total, 651 sentences of the corpus, 15,140 tokens in total, were manually corrected, out of 80,395 tokens overall. Two annotators with a background in linguistics worked on the manual correction. Sentences were corrected to adhere to annotation rules used in the Icelandic Parallel Universal Dependencies (PUD) corpus (Jónsdóttir and Ingason, 2020), which is the only Icelandic UD corpus which was created manually. The corpus was used as a guideline when UDConverter was developed. The annotators worked on separate sentences, and therefore information on inter-annotator agreement is not available. It would be beneficial to have information on the agreement, but the annotators discussed any uncertainties and came to joint conclusions.

We used a labeled attachment score (LAS) to evaluate the converter, evaluating CoNLL-U output based on how many tokens have been assigned both the correct syntactic head and the correct dependency relation (Kübler et al., 2009). This simple accuracy score corresponds to a labeled $F_1$ score of syntactic relations. Similar to this score is the unlabeled attachment score (UAS), which evaluates the number of correct heads but does not take the dependency relations into account.

## 4 Results

Our results show that the converter achieves an LAS of 82.87 and a UAS of 87.91. Our results indicate that the overall error rate of the conversion is not affected by sentence length, with the relationship between sentence length and total errors per sentence being more or less linear. If sentence length is a rough indicator of syntactic complexity, this means that the converter handles complex syntactic structures just as well as simpler ones. This is expected, as the converter works off of a fixed rule set for a given language, which looks at the already annotated phrase structure of the input sentences.

### 4.1 Initial results

The first evaluation of the converter showed worse results, with an LAS of 72.82 and a UAS of 80.79. After analyzing the difference in the converter's output and the manually corrected texts, a few systematic errors were identified, which accounted for a large proportion of errors. Three of these items related to an incorrect head of a dependent with a particular dependency relation, and two related to an incorrect dependency relation.

**Head-related errors**

The three head-related errors have the dependency relations *punct*, *cop* and *cc*. *Punct* is used to denote punctuation and was dependent on an incorrect head in 75.63% of cases. An important error relating to *punct* was in the case of end-of-sentence punctuation, which should be dependent on the root of the sentence. 66.28% of *punct* dependency relations dependent on an incorrect head were end-of-

sentence punctuation, i.e. punctuation marks which should have been dependent on the sentence's root, but were for some reason not.

The second head-related error was the *cop* dependency relation, with a 21.86% error rate. This relation is used for copulas, which in Icelandic is the verb *vera* 'be'. Copular constructions are structurally different from other verbal constructions, so this construction had to be handled specifically, marking the predicate as the root of a sentence and the copular verb as its dependent. Determining which word or phrase is the predicate is not always unequivocal, so a copular verb is in some cases dependent on the incorrect word.

The third and final head-related error was the *cc* dependency relation, which was dependent on an incorrect head in 18.52% of cases. This relation is used for a coordinating conjunction and is part of a conjunction phrase in IcePaHC. In a simple example, a conjunction phrase is made up of three words, e.g. two nouns with a coordinating conjunction between them, linking them together. Initially, the converter marks the first noun as the head of the phrase and the conjunction and the second noun as its dependents. According to the UD annotation guidelines, the conjunction should be dependent on the second noun, so this is corrected in the conversion algorithm as part of a series of checks after the initial conversion is done, making the second noun the head of the conjunction. In more complex cases, this correction can go wrong, resulting in the conjunction (*cc*) being dependent on an incorrect head.

**Incorrect dependency relations**

The two most frequent incorrect dependency relations were *acl* and *obl*. The *acl* relation stands for finite and non-finite clauses that modify a nominal. It had an error rate of 72.01% and was, in most cases, supposed to be replaced by the *xcomp* relation, which denotes an open clausal complement of a verb or an adjective. This error was caused by a fault in the rules of UDConverter, wherein the *acl* relation was incorrectly used for heads of certain subcategories of infinitival clauses, e.g. direct speech, degree infinitives and subjectival infinitives. These clauses are labeled *IP-INF* in the IcePaHC annotation scheme, and this relation was incorrectly mapped to *acl* instead of *xcomp*. These errors were therefore simple to correct.

The second incorrect dependency relation, *obl*, had an error rate of 26.44%. The *obl* relation is

used for a nominal which functions as an oblique argument or adjunct. A proportion of these errors are due to the fact that the *obl:arg* relation is used in the manually corrected sentences, but not in the converter. *obl:arg* is a subcategory of the *obl* relation, and is used to distinguish oblique arguments from adjuncts, which have the *obl* relation. This relation was used to have our manually corrected sentences better conform to the Icelandic PUD corpus, which uses this relation.

These five items were analyzed, e.g. how often a relation which should have been *xcomp* was incorrectly *acl*, and a projection was created on the converter's possible LAS if these errors were fixed altogether. This projected LAS is 85.34, which is considerably higher than the original 72.82.

## 4.2 Final results

After having analyzed the improvements discussed in Section 4.1, most were updated in UDConverter. The only improvement not added was including *obl:arg* as a possible dependency relation. The difference between *obl* and *obl:arg* is semantic, and it is not accounted for in IcePaHC sentences. It therefore proved complicated to add the relation to the converter, and external information would have to be obtained in order for *obl:arg* to be used.

The four other types of errors discussed above were improved, resulting in error rates shown in Table 1. Rules regarding heads of end-of-sentence punctuation were improved, and the resulting error rate is 29.03%. Rules on head selection of copular verbs were improved by examining individual errors, which resulted in a 7.99% error rate. Head selection of the *cc* dependency relation was also improved, again by examining individual occurrences and adding to the converter's rules. This resulted in a 3.70% error rate. The final improvement made to UDConverter was to the *acl* dependency relation. As discussed in Section 4.1, this error was simple to correct, and rules in the converter were updated to account for this, resulting in a 31.25% error rate.

As discussed, these improvements resulted in the current LAS of 82.87 and UAS of 87.91. These accuracy scores are not consistent with the projected LAS of 85.34, which assumes that all error instances are handled and that the *obl:arg* dependency relation is added to the converter. Nevertheless, the error rates drop considerably, the LAS increasing by 10.05 points and the UAS by 7.12 points. These accuracy scores were obtained by

| Deprel to fix | Prev. error rate | Final error rate |
|---|---|---|
| punct | 75.63% | 29.03% |
| cop | 21.86% | 7.99% |
| cc | 18.52% | 3.70% |
| acl | 72.01% | 31.25% |
| obl:arg | 27.73% | 27.73% |

Table 1: Dependency relations associated with errors in the converter along with the converter's possible LAS after being improved, with respective score gain.

measuring on the same test set as the one that was used for initial evaluation. This method presents some limitations and can cause a bias in the results. The improvements to the converter might be overfitted on the test set, resulting in higher accuracy scores. To counteract this, a development set must be created, manually correcting more sentences and using them to obtain updated accuracy scores.

## 5 Comparison

Various automatic methods are available to create a UD corpus for Icelandic. To determine the most beneficial method of creating Icelandic UD corpora, we compare UDConverter's accuracy scores to three UD parsers: a UDPipe 1 (Straka and Straková, 2017) model specifically trained to be compared to UDConverter, and the two parsers discussed in Section 2; the Diaparser-based one and the COMBO-based one.

Our UDPipe model was trained on the converted UD_Icelandic-IcePaHC and was used to parse the same sentences as the manually corrected parliament speeches, which were then compared to our manual corrections. While the model tags correctly 92.87% of the time using the Universal Dependencies tagset (UPOS) and 86.78% of the time with the IcePaHC tagset (XPOS), the LAS is only 55.29 and UAS 63.03, which is substantially lower than the output of our converter. Using the same test set, we measured the accuracy of the Diaparser-based parser and the COMBO-based parser. Diaparser delivers a 71.46 LAS and a 78.29 UAS, and the COMBO-based one delivers a 71.04 LAS and a 77.71 UAS. These accuracy scores, in comparison to the scores for UDConverter, are shown in Table 2.

All three parsers, which are the only available Icelandic UD parsers, are trained using output from the converter, which presents some limita-

| Method | LAS | UAS |
|---|---|---|
| UDPipe | 55.29 | 63.03 |
| Diaparser | 71.46 | 78.29 |
| Combo-parser | 71.04 | 77.71 |
| UDConverter | 82.87 | 87.91 |

Table 2: Accuracy scores of the parsers as compared to UDConverter.

tions when comparing them to the converter. The parsers learn from the training data, and can never produce results which are as accurate as the data itself. Comparing the parsers' output to the converter's output is therefore not an equal comparison, but it does give an idea about their accuracy. Furthermore, accuracy scores for UDConverter are possibly higher than if they were obtained from development data, as discussed above. Current scores show that using UDConverter to create UD corpora will deliver the most accurate results, as the highest accuracy score for the three parsers is 11.41 points less than the converter's accuracy. However, each method has its advantages and drawbacks, as a converter requires a treebank which is annotated in the appropriate format, while parsers can create a corpus from plain text.

## 6 Conclusion

We have described the evaluation of a rule-based conversion tool, UDConverter, which converts treebanks in the phrase-structured PPCHE format to the dependency-based UD format. Converted texts were manually corrected and used as testing data. We focused on the accuracy of dependency heads and dependency relations to achieve labeled and unlabeled accuracy scores (LAS, UAS), which serve as F1 scores in our evaluation.

Our results show that UDConverter achieves an LAS of 82.87 and a UAS of 87.91. We compared these accuracy results to accuracy scores of three different Icelandic UD parsers, our UDPipe model along with Diaparser and Combo-parser, which showed that using UDConverter most accurately delivers an Icelandic UD corpus.

## Acknowledgements

This project was funded by the Language Technology Programme for Icelandic 2019–2023. The programme, which is managed and coordinated by Almannarómur (https://almannaromur.is/), is funded by the Icelandic Ministry of Education, Science and

Culture. We would like to thank the anonymous reviewers for their contribution.

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
