# OpenReview forum: "Evaluating a Universal Dependencies Conversion Pipeline for Icelandic"
_NoDaLiDa/2023/Conference — NoDaLiDa 2023_

### Official Review · Reviewer_wUfp · 2023-03-03
**Evaluation of the UDConverter for Icelandic using manually corrected data, with good results**

**Rating:** 7
**Confidence:** 4

**Review:**

The paper deals with the tool UDConverter, which can convert constituent parses into UD dependencies. The goal of the paper is an evaluation of the correctness of the converter. For this purpose, the CoNLL-U output format of the converter is manually checked and corrected using 651 sentences (15K tokens). The converter achieves scores of 82.87 (LAS) and 87.91 (UAS). A detailed error analysis reveals a number of errors in the converter, which are subsequently corrected (and then achieve the above scores). Additionally, the converted parses are compared with automatically generated parses of 3 parsers, which give worse results (but were trained on the converted data only).

Pros:
* Important work to evaluate the correctness of the converter to be able to estimate the quality of the resulting converted treebanks.
* The paper is well written and easy to follow.

Cons:
* There is no information on how the manual corrections were done. According to the paper, two annotators were involved, but it is not said whether they corrected independently and if so, what was the level of agreement between them (IAA). This information would be important to estimate the quality of the test set (which in turn is used to check the quality of the converter).
* After a first evaluation, the converter was improved again and then re-evaluated on the same test set. Such a procedure is actually not correct, since the test set must consist only of unseen data. (The problem is at least addressed in the paper).

**Paper Type:**

Short paper

---

### Official Review · Reviewer_Wda9 · 2023-03-06
**-**

**Rating:** 8
**Confidence:** 5

**Review:**

The paper reports on the development and evaluation of the UD converter for Icelandic. The paper is clearly written and easy to read. The contribution is certainly useful for the development of parser for Icelandic. In the approach, however, there is nothing fundamental - the problems reported are quite trivial and easy to solve once an evaluation is done. I would have expected that this kind of evaluation should have been done during the initial construction of the tool. Given that I recommend rate 8 rather than some of the higher scores.

**Paper Type:**

Short paper

---

### Official Review · Reviewer_jPf7 · 2023-03-08
**Useful tool of limited general interest and few general conclusions.**

**Rating:** 6
**Confidence:** 4

**Review:**

This paper presents an evaluation of a tool for converting phrase structure treebanks for Icelandic in the format of Penn Parsed Corpora of Historical English to dependency treebanks in the Universal Dependencies treebank. This is undoubtedly a useful tool for researchers working on Icelandic treebanks (in the specific formats), but it is of very limited interest to the wider community, especially since the evaluation is presented in such a way that it is hard to draw any general conclusions that we can learn from when converting treebanks for other languages or in other formats. It would add to the general interest of the paper if the authors could include a discussion about how the results could be generalized, or about how the tool could be used for other languages. Apparently, the tool has been used not only for Icelandic but also for Faroese, but this is only mentioned in passing in the paper.

Specific comments:

In the introduction, it is stated that “[w]ithout such evaluation, the benefit of using the converter is uncertain”. What exactly is meant by this? Since it is only a conversion tool, it can be useful even if we don’t know exactly how accurate it is, can it not?

In the background section, UDconverter is described as a tool for “converting bracket-parsed treebanks in the format of the Penn Parsed Corpora of Historical English (PPCHE) to the Universal Dependencies framework”. This seems to suggest that the tool is more widely applicable and could be used also, for example, for English treebanks, but this is never discussed in the paper.

I don’t understand how the parsers described in the last paragraph of the background section are relevant for the current paper.

Comparing the UAS/LAS scores for the converter to those obtained with syntactic parsers is a nice addition that suggests that it is better to use the converter than a parser as pre-processor to manual annotation/editing (although parsers of course have the advantage that they can be applied to data that has not already been annotated in another framework). Another experiment that could be considered is to investigate whether a parser trained on the output of the improved converter gets higher accuracy than one trained on the output of a previous version.


**Paper Type:**

Short paper

---

### Decision · Program_Chairs · 2023-03-17

Accept